# Response Gene to Complement 32 promotes cell proliferation and tamoxifen resistance in breast cancer via elevated FoxM1 expression

Xinlei Li[1☉], Yan Liu[2☉], Zhiqian Wang[3☉], Xiaocui Bu[4], Yu Wang[1], Wei Zhang[2*], Peng Zhao[3*]

1 Medical College of Qingdao University, Qingdao, China, 2 Department of Pathology, The 971 Hospital of People's Liberation Army Navy, Qingdao, China, 3 Department of Molecular Pathology, Qingdao Central Hospital, University of Health and Rehabilitation Sciences, Qingdao, China, 4 The Affiliated Cardiovascular Hospital of Qingdao University, Qingdao, China

☉ These authors contributed equally to this work.
* zp8102@126.com (PZ); zhangwei686538@126.com (WZ)

## Abstract

Despite the high sensitivity of estrogen receptor positive (ER⁺) breast cancer to endocrine therapy, many patients have primary resistance or develop resistance to endocrine therapies. Acquired resistance to endocrine therapy is a great challenge in the treatment of ER⁺ breast cancer patient. Here we showed that Response Gene to Complement (RGC)-32 expression is higher in breast cancer than paired normal tissues, which was a poor predictive factor. RGC-32 overexpression resulted in tamoxifen resistance, whereas knockdown of RGC-32 in tamoxifen-resistant cells restored tamoxifen sensitivity. Tamoxifen resistance mediated by RGC-32 was shown to be partially dependent on FoxM1 expression. Mechanistically, RGC-32 could activated PI3K signaling pathway, and then enhanced estrogen receptor alpha (ERα) activity. ERα activation is essential for RGC-32-mediated the expression of FoxM1. These data support that targeting RGC-32 could effectively mitigate cancer progression and tamoxifen resistance, offering a complementary therapeutic approach to reduce acquired endocrine resistance.

## Introduction

Breast cancer is the most common female cancer types worldwide [1]. Up to 70% of patients with breast cancer are estrogen receptor positive (ER⁺), and its growth and proliferation were controlled by estrogen receptor signaling pathway [2]. Tamoxifen, an ER alpha modulator, has been the standard first-line endocrine therapy for early and advanced breast cancer patients in the past 30 years [3]. However, the development of tamoxifen resistance is common, making it an outstanding problem in breast cancer therapy [4,5]. Although tremendous efforts have been made to understand the

**Data availability statement:** All relevant data are within the manuscript and its Supporting information files.

**Funding:** This work was supported by a grant from the National Natural Science Foundation of China (NO. 82072927). The funder had no role in study design, data collection and analysis, decision to publish, or preparation of the manuscript.

**Competing interests:** The authors have declared that no competing interests exist.

mechanism underlying tamoxifen resistance, the detailed molecular mechanism are unclear.

Response Gene to Complement (RGC)-32 is initially identified as a complement activation-inducible gene that promotes cell cycle progression [6]. It has been also found to be abnormally expressed in a variety of cancers, and plays a dual role in cancer, functioning as either a tumor promoter by enhancing tumor cell proliferation, invasion, metastasis, and angiogenesis, or as a tumor suppressor [7]. RGC-32 expression is upregulated in EBV-driven B cell transformation and disrupts the G2/M checkpoint via CDK1 activation [8]. In addition, RGC-32 serves as a hypoxia-responsive regulator involved in epithelial-mesenchymal transition of pancreatic cancer [9]. Conversely, overexpression of RGC-32 in gliomas inhibits the growth of tumor cells, which is directly induced by p53 and leads to mitotic arrest [10]. Notably, upregulation of RGC-32 has been frequently observed in breast cancer but its functions and regulatory mechanisms remain unexplored [11,12].

In the study, we systematically investigate the functional roles of RGC-32 in ER⁺ breast cancer. We demonstrated that higher RGC-32 expression promoted tumor growth and acquired tamoxifen resistance in ER⁺ breast cancer. These finding suggests that identifying ways to inhibit RGC-32 expression may be a therapeutic target in ERα⁺ breast cancer patients who resistant to endocrine therapies.

## Results

### RGC-32 expression is associated with poor prognosis in ER⁺ breast cancer

Since RGC-32 expression is regulated by steroid hormones [13,14], we performed immunohistochemistry on the collected tissues to examine the expression pattern of RGC-32 in ER⁺ breast cancer. Our results showed that RGC-32 expression was higher in ER⁺ breast cancer compared with adjacent non-tumor tissues (Fig 1A, B). We also found that RGC-32 expression in stage I-II was notably higher than that in stage III-IV (Fig 1C). To further investigate the effect of RGC-32 expression on cancer progression, we analyzed the correlations between the expression of RGC-32 and clinicopathologic parameters in patients with ER⁺ breast cancer. Table 1 showed that high RGC-32 expression was correlated with TNM stages and Ki67 expression in ER⁺ breast cancer (Table 1). Furthermore, survival analysis indicated that RGC-32 low group had a significant survival advantage compared with RGC-32 high group (Fig 1D). The univariate and multivariate Cox regression analysis revealed that RGC-32 prognostic score was an independent prognostic predictor for overall survival (Table 2).

### RGC-32 is a tumor promoter in ER⁺ breast cancer cells

For functional studies, the influence of inhibition or forced expression of RGC-32 were performed in ER⁺ breast cancer cells (MCF-7 and T-47D). Lentivirus vector containing RGC-32-small hairpin RNA (shRNA) or non-targeting vector-control shRNA or plasmids harboring RGC-32 sequences was stably transfected into ER⁺ breast cancer cell lines. The expression level of RGC-32 in these cells were verified by

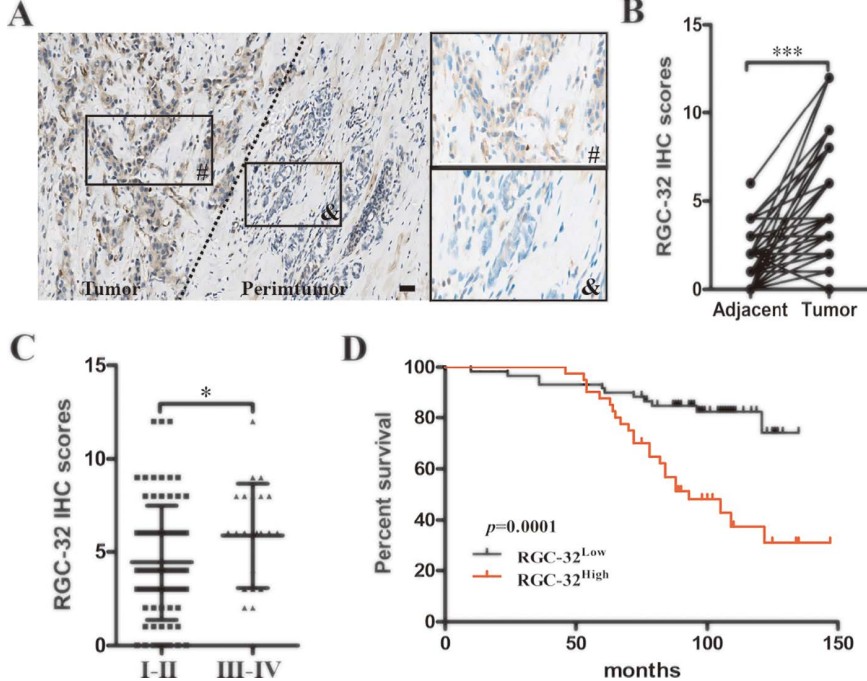

**Fig 1. RGC-32 expression is corelated with poor survival in ER+ breast cancer patients.** A, Representative images of RGC-32 expression in peritumoral and tumor tissue in ER+ breast cancer. Scale bar = 100 μm. B, Staining scores for RGC-32 expression in the adjacent non-tumor tissues and ER+ breast cancer. C, RGC-32 expression levels in ER+ breast cancer patients with stage I-II and stage III-IV. D, Comparison between of the survival of ER+ breast cancer patients with low or high expression level of RGC-32. *P < 0.05; **P < 0.01; ***P < 0.001.

western-blot (S1A, B Fig). A growth curve assay showed that RGC-32 knockdown markedly reduced the growth of breast cancer cells (Fig 2A), while RGC-32 overexpression promoted cell growth (Fig 2B). Concordantly, RGC-32 knockdown increased G0/G1 phase fractions and decreased the percentages of abnormal mitosis cells, while RGC-32 overexpression showed the opposite effects (Fig 2C, D). These results indicate that RGC-32 acts as an oncogene in ER+ breast cancer.

### RGC-32 reduces the sensitivity of ER+ breast cancer cells to tamoxifen

Next, we further studied whether RGC-32 regulated the sensitivity and acquired resistance of ER+ breast cancer cells to tamoxifen. Our results showed that the effect of tamoxifen on cell viability was compromised when RGC-32 was over-expressed in MCF-7 and T-47D cells (Fig 3A, C). RGC-32-overexpressing cells exhibited a smaller tamoxifen-mediated increase in G0/G1 phase fraction compared with control cells (Fig 3B, D). In addition, we generated an acquired resistant cellular model, MCF-7 TamR cells (S1C Fig). We found that RGC-32 expression was higher in TamR cells than that in parental cells (S1D Fig). TamR cells were transfected with RGC-32-shRNA vector and treated with tamoxifen (S1E Fig). Our results showed that RGC-32 knockdown resulted in more sensitivity to tamoxifen, indicative of partial re-sensitization towards tamoxifen (Fig 3E).

### RGC-32 mediates tamoxifen resistance through increased FoxM1 expression

To elucidate the molecular mechanism(s) utilized by RGC-32 in breast cancer cells, we further performed a microarray analysis of gene expression in RGC-32 knockdown and control MCF-7 cells. The microarray revealed that 452 genes increased, but 446 genes decreased in RGC-32 knockdown cells compared with the control group (Fig 4A). We detected

**Table 1. RGC32 expression in relation to clinicopathological parameters of patients with ER⁺ breast cancer.**

| Clinical characteristics | RGC-32 | | P value |
|---|---|---|---|
| | Low | High | |
| **Age** | | | 0.946 |
| **<60** | 35 | 24 | |
| **≧60** | 24 | 16 | |
| **Grade** | | | 0.434 |
| **1** | 4 | 2 | |
| **2** | 15 | 15 | |
| **3** | 40 | 23 | |
| **Tumor size** | | | 0.268 |
| **< 5 cm** | 52 | 32 | |
| **≧ 5 cm** | 7 | 8 | |
| **Lymph node** | | | 0.082 |
| **0** | 37 | 18 | |
| **1** | 22 | 22 | |
| **TNM stage** | | | 0.02 |
| **I-II** | 51 | 26 | |
| **III-IV** | 9 | 14 | |
| **P53 expression** | | | 0.461 |
| **Negative** | 34 | 26 | |
| **Positive** | 25 | 14 | |
| **Ki67** | | | 0.035 |
| **<10%** | 16 | 8 | |
| **10%−30%** | 32 | 15 | |
| **>30%** | 11 | 17 | |

**Table 2. Multivariate analyses of prognostic factors correlated with OS.**

| Variable | Univariable | | | Multivariate | | |
|---|---|---|---|---|---|---|
| | HR | 95% CI | p value | HR | 95% CI | p value |
| Grade (1/2/3) | 1.328 | 0.713-2.473 | 0.371 | 1.544 | 0.81-2.945 | 0.187 |
| TNM stage (I -II/III-IV) | 3.71 | 1.815-7.583 | 0 | 2.65 | 1.233-5.695 | 0.013 |
| RGC-32(low/high) | 3.737 | 1.75-7.978 | 0.001 | 2.722 | 1.183-6.263 | 0.018 |
| Ki67 expression (<10/10–30/≧30) | 1.349 | 0.812-2.242 | 0.248 | 1.054 | 0.629-1.766 | 0.843 |
| Tumor size (<5/≧5) | 2.265 | 1.006-5.097 | 0.048 | 1.672 | 0.737-3.795 | 0.219 |

a signature by recruiting several probes with a cutoff value of >1.3 fold change in RGC-32-knockdown cells compared with control cells. Cell cycle control of chromosomal replication was predicted to the top pathway whose members exhibited gene expression changes after RGC-32 knockdown according to Ingenuity Pathway Analysis (IPA). In an upstream pathway analysis using transcriptome data, estrogen-mediated S phase entry was one of the top inhibited pathways after RGC-32 silencing (Fig 4B).

FOXM1 is a key mediator of mitogenic functions of ERα and estrogen in breast cancer cells, which plays a pivotal role in promoting cell survival and resistance to endocrine therapies [15,16]. We found that inhibition of RGC-32 affected the expression of FoxM1 and its known target genes (Fig 4C). Consistent with the microarray data, MCF-7 and T-47D cells transfected with RGC-32 shRNA exhibited decreased FoxM1 expression compared with their control cells (Fig 4D). SKP2

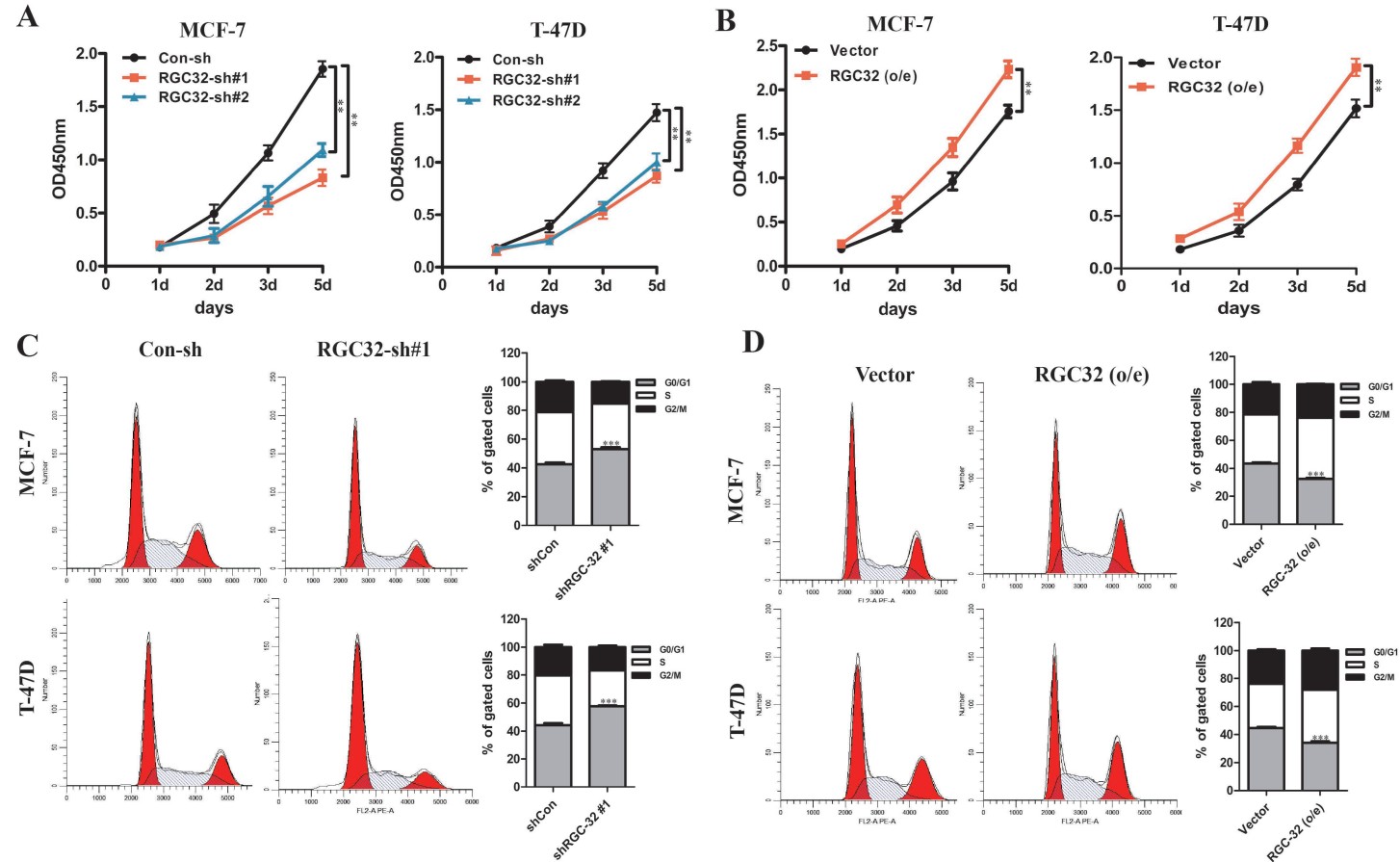

**Fig 2. RGC-32 expression increases cell proliferation in ER+ breast cancer cells.** A, Cell viabilities of breast cancer cells transfected with vectors targeting RGC-32 or non-specific sequence were determined by CCK8 assay over a period of 5 days. B, Cell proliferation in breast cancer cells with forced expression of RGC-32 or control vector was determined. C-D, Cell cycles of the respective breast cancer cells were analyzed by flow cytometry.

and SOX2 mRNA expression were also reduced in RGC-32 knockdown cells (S2A, B Fig). In addition, we found that RGC-32 expression was positively correlated with the expression of FoxM1 in breast cancer patients (Fig 4E, F). To verify whether FoxM1 mediated the functionality of RGC-32, we construct FoxM1 overexpression vector (S2C Fig). Our results showed that FoxM1 overexpression reversed the inhibitory effects of RGC-32 knockdown on the expression of SKP2 and SOX2 (S2D, E Fig), as well as cell viability (Fig 4G). In addition, we found that FoxM1 overexpression partially reduced the sensitivity of RGC-32 knockdown TamR cells to tamoxifen compared to their respective cells (Fig 4H).

### RGC-32 increases FoxM1 expression in breast cancer cells through PI3K- ERα crosstalk

The crosstalk between ERα and elevated PI3K/AKT activity has been reported to be pivotal for tamoxifen resistance in breast cancer [17,18]. Our results showed that the phosphorylation of AKT and ERα was enhanced in TamR cells (Fig 5A). The activation of PI3K/AKT pathway is modulated by RGC-32 [19,20]. In coincidence, the phosphorylation of AKT was increased in RGC-32 overexpression cells, accompanied by elevated ERα phosphorylation (Fig 5B). We found that inhibition of PI3K signaling by inhibitor LY290042 attenuated ERα phosphorylation mediated by RGC-32 overexpression, suggested RGC-32 regulated ERα activation through PI3K signaling (Fig 5C). We conducted an estrogen response element

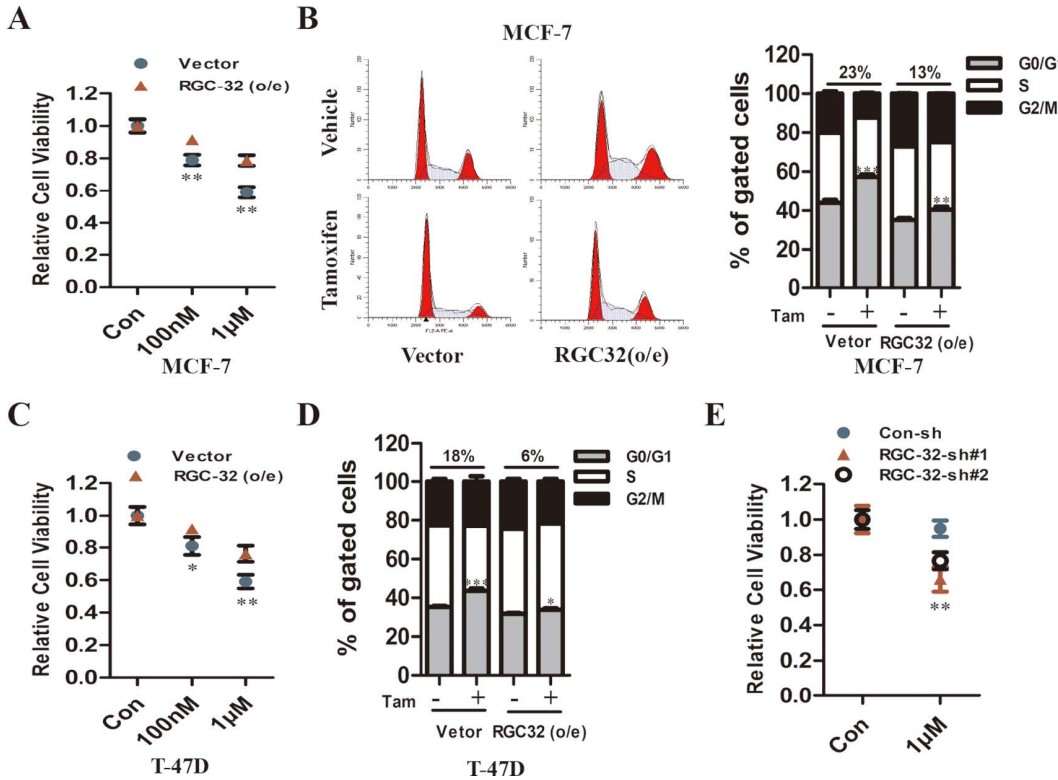

**Fig 3. RGC-32 reduce tamoxifen sensitivity in ER⁺ breast cancer cells.** A and C, RGC-32 overexpressing and vector cells were treated with vehicle control (DMSO), 100nM and 1μM tamoxifen for 5 days. Cell viability was determined by CCK8 assay. Values were normalized to control within each cell line and are the mean±SEM from 3 independent CCK8 assays. B and D, Cell cycles were analyzed by flow cytometry. E, MCF-7 TamR cells stably transfected with RGC-32 shRNA and control vectors were treated with 1μM tamoxifen for 5 days. Cell viability was determined by CCK8 assay.

(ERE) luciferase assay in breast cancer cells to investigate the impact of RGC-32 overexpression on ERα transcriptional activity. Our results showed that overexpression of RGC-32 could enhance the transcription activity of ERE (Fig 5D).

FOXM1 is a key mediator of mitogenic functions of ERα and estrogen in breast cancer cells. To determine whether the increase in FOXM1 expression observed in response to RGC-32 overexpression was caused by ERα, the cells were treated with fulvestrant (ICI) to degrade the ERα. Our results showed that ICI blocked the promoting effect of RGC-32 on FoxM1 expression, suggesting that FoxM1 expression is estrogen-ERα dependent (Fig 5E). The activation of Cdk1 involves the removal of inhibitory phosphorylation at Thr-14 and Tyr-15, which led to phosphorylation of FOXM1 [21,22]. Our results revealed that RGC-32 overexpression resulted in a decrease in Cdk1-Thr14/Tyr15 phosphorylation (Fig 5F). To determine whether the increase in p-FOXM1 observed in response to RGC-32 overexpression was caused by Cdk1, plasmids containing Cdk1-small hairpin RNA was transfected into ER⁺ breast cancer cell lines. The results showed that the positive effect of RGC-32 overexpression on FoxM1 phosphorylation (Thr 600 and Ser 35) was blocked by Cdk1 knockdown (Fig 5G).

## RGC-32 overexpression increases FoxM1 and Ki67 expression in the xenograft and partially reduced the sensitivity to tamoxifen

To confirm whether RGC-32 expression alter the sensitivity of breast cancer to tamoxifen in vivo, we established a tumor xenograft model by subcutaneously implanting breast cancer cells expressing either control or RGC-32 vector into athymic nude mice. Mice were divided into four groups (control, RGC-32 (o/e), control+TAM, RGC-32+TAM). After tamoxifen

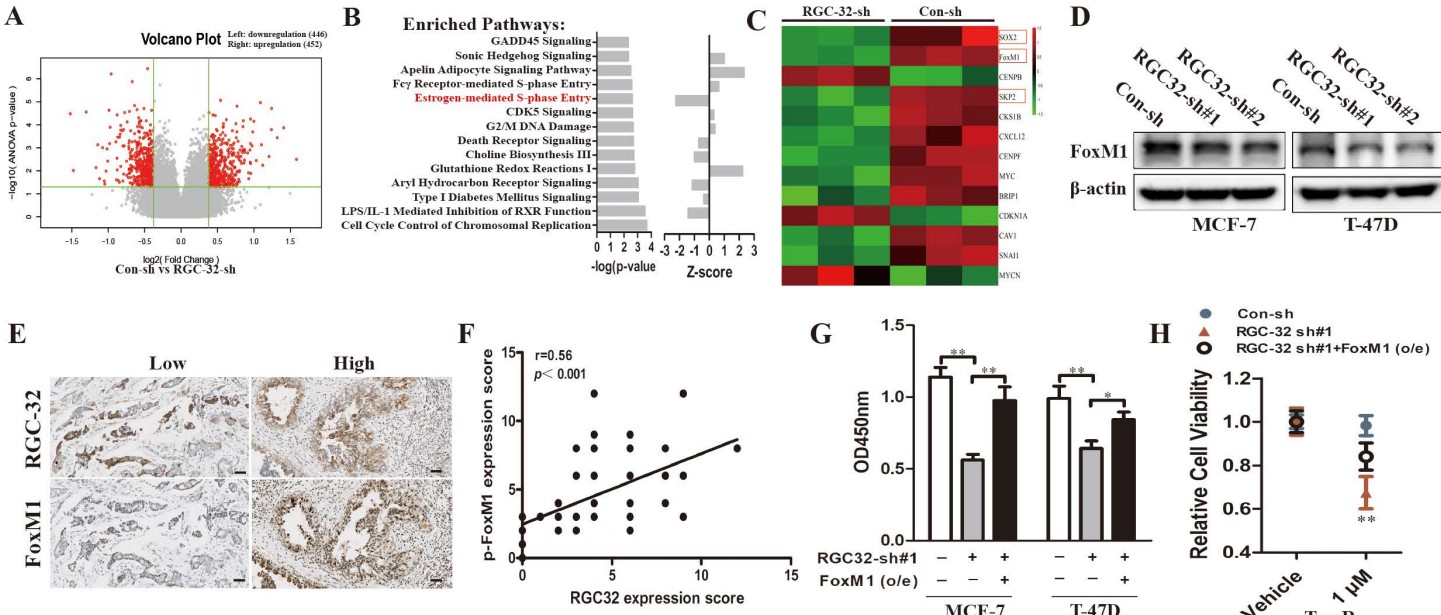

**Fig 4. Identification of RGC-32 target in breast cancer.** A-C, Volcano plots of the differential gene expression in RGC-32 knockdown and control cells. B, Top-ranked cellular pathway classification of the DEGs in control vs. RGC32 knockdown MCF-7 cells analyzed using Ingenuity Pathway Analysis (IPA). C, Heatmap of relationship of RGC-32 and FoxM1 related target genes. D, Breast cancer cells were stably transfected with shRGC-32 or NC. FoxM1 expression were determined by western-blotting. E, Immunohistochemistry of selected cases with low (upper) and high (bottom) expression of RGC-32 and phospho-FoxM1 (Ser35) in ER⁺ breast cancer. Scale bar = 100 μm. F, The correlation between RGC-32 and phospho-FoxM1 IHC scores in ER⁺ breast cancer cells was evaluated. G, Breast cancer cells were co-transfected with plasmids expressing FoxM1, RGC-32-shRNA or empty vector. Cell viability was determined by CCK8 assay. H, TamR cells were co-transfected with plasmids expressing FoxM1, RGC-32-shRNA or empty vector. These cells were treated with 1μM tamoxifen for 5d. Cell viability was determined by CCK8 assay. *P<0.05; **P<0.01; ***P<0.001.

treatment, the growth of tumors derived from MCF-7 cells overexpressing RGC-32 exhibited a greater tumor weight compared with control group, suggesting that RGC-32 overexpression decreased tamoxifen sensitivity in vivo (Fig 6A–C). Moreover, IHC results revealed that MCF-7 cells overexpressing RGC-32 showed higher FoxM1 and Ki67 expression compared with control groups, with or without tamoxifen (Fig 6D).

## Discussion

Tamoxifen, an estrogen selective modulator is widely used standard-of-care treatment for ER⁺ breast cancer patients. Despite its efficacy, the development of resistance to tamoxifen represents a significant obstacle in the treatment of breast cancer. Therefore, elucidating the molecular mechanisms of acquired drug resistance in breast cancer represents an urgent unmet medical necessity. In the study, we demonstrated that RGC-32 overexpression reduced the sensitization of breast cancer cells to tamoxifen. Conversely, inhibition of RGC-32 resulted in the partial re-sensitization of tamoxifen-resistant cells. These data suggested a role for RGC-32 in tamoxifen resistance.

Previous studies have shown that RGC-32 expression is significantly upregulated in adenocarcinoma and adenoma compared to normal colon mucosa, with higher expression levels observed in advanced-stage patients relative to early-stage cases [23]. Consistent with these results, we observed that RGC-32 is overexpressed in ER⁺ breast cancer compared with adjacent non tumor tissues, and predicted poor prognosis. In vitro studies confirmed that RGC-32 promoted the proliferation of ER⁺ breast cancer cells, suggestive of a promoting tumor role of RGC-32. To further explore the network of proteins regulated by RGC-32, we took a microarray analysis to identify genes that are differentially expressed

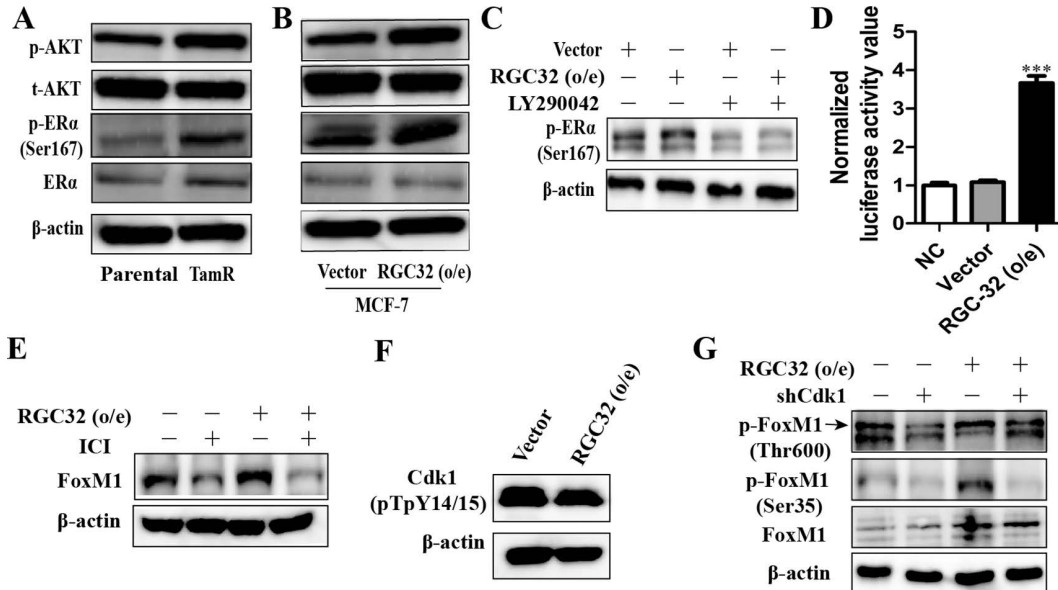

**Fig 5. RGC-32 augments PI3K/AKT-ER αcrosstalk to increase FoxM1 expression in ERα+ breast cancer cells.** A-B, The levels of p-ERα (Ser167), p-AKT, ERα and total AKT in the indicated cells were analyzed by western blot. C, RGC-32 overexpressing and control cells were treated with 10μM LY294002. The levels of p-ERα (Ser167) and ERα expression were determined by western-blotting. D, ERE-luciferase activity was detected by luciferase assays in MCF-7 cells. E, ER⁺ breast cancer cells transfected with plasmids containing RGC-32, or empty vector were treated with/without 1μM ICI 182,780. FoxM1 expression were determined by western-blotting. F, The protein level of Cdk1 (pTpY14/15) were analyzed in ER⁺ breast cancer cells transfected with plasmids containing RGC-32, or empty vector. G, RGC-32 overexpressing or control cells was transfected with shCdk1. The phosphorylation levels of FoxM1 were determined by western-blotting.

after RGC-32 silencing. RGC-32 has significant impact on multiple signaling pathways involved in regulating the cell cycle, such as cell cycle control of chromosomal replication, estrogen-mediated S phase entry and Fc-γ Receptor-mediated S-phase Entry. Sonia I. Vlaicu et al. have found that the genes implicated in chromatin assembly, cell cycle, and RNA processing were significantly regulated by RGC-32 in colon cancer [23]. These results suggest that RGC-32 cooperates with multiple proteins, including those of oncoproteins and tumor suppressor proteins to promote or suppress tumor growth in different cell lines, which may be reason for the observed incongruity in the effects of RGC-32 in different cell lines [24–26].

RGC-32 forms a functional complex with Cdk1 and enhances its kinase activity, thereby accelerating S-phase entry. This kinase-enhancing activity requires phosphorylation of RGC-32 at threonine 91 (Thr91), a critical modification mediated by Cdk1 [6]. The activation of both ERK1, a member of the mitogen-activated protein kinase (MAPK) family, and PI-3K pathway contribute to cell-cycle induction by C5b-9 [27]. Notably, RGC-32 serves as a substrate for ERK1-mediated phosphorylation at Thr91 [28]. Additionally, RGC-32 can bind to and was phosphorylated by AKT at Ser-473 [29]. Simultaneously, RGC-32 can induce the phosphorylation of AKT to promote cytokine transcription, survival, and cell-cycle activation [30,31]. Our results showed that the phosphorylation of AKT is significantly increased in RGC-32-overexpressing cells. It has been well established that ERα can be phosphorylated by PI3K/AKT, and phosphorylated ERα has been implicated to regulate ligand-independent activation of ERα and tamoxifen resistance [32–34]. Consistent with these reports, inhibition of PI3K/AKT pathway significantly reduced the activation of ERα mediated by RGC-32 overexpression. These results suggested a regulatory relationship between RGC-32 and PI3K/AKT-ERα crosstalk.

FOXM1, a transcription factor of the Forkhead family, plays a crucial role in regulating cell cycle progression, DNA damage repair, and maintaining genomic stability [35]. It is frequently overexpressed in multiple tumor types, including ovarian,

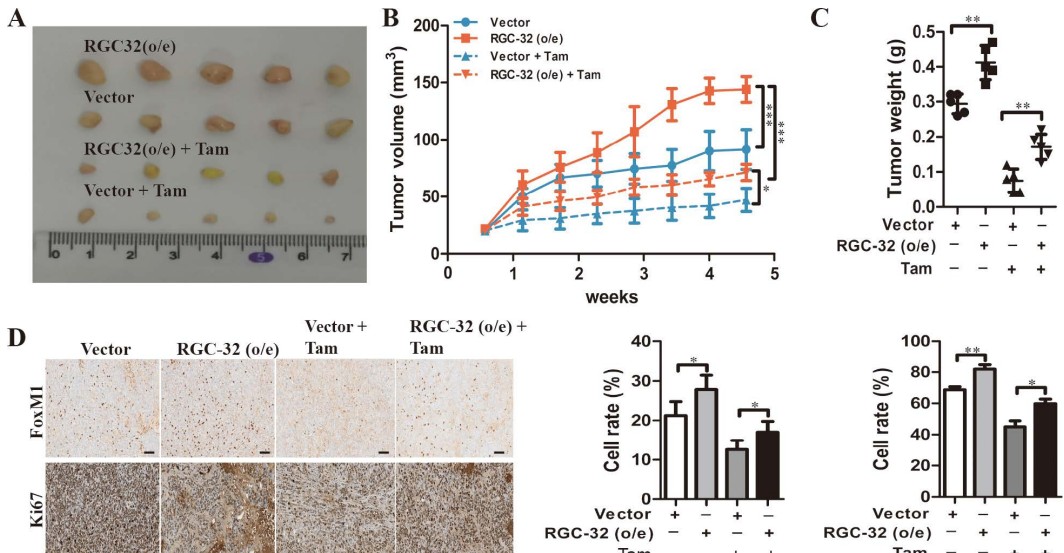

**Fig 6. RGC-32 reduces tamoxifen sensitivity of ER⁺ breast cancer cells in vivo.** A-C, MCF-7 cells stably transfected with RGC-32 and control vectors were injected into the mammary fat pads of nude mice. The mice were treated with tamoxifen in the presence of an exogenous estrogen supplement. Representative tumor pictures, xenograft tumor growth curves and tumor weight were shown. D, Immunohistochemical analysis of Ki67 and p-FoxM1 expression in xenograft tumors. Scale bar = 100 μm. *$P < 0.05$; **$P < 0.01$; ***$P < 0.001$. E, A working model showing that RGC-32-ERα-FoxM1 axis contributes to tumor growth and tamoxifen resistance.

colorectal, esophageal, breast, prostate, gastric, and pancreatic cancers [36]. Clinically, elevated FoxM1 expression is associated with poor prognosis and development of therapeutic resistance, thereby establishing FOXM1 as a promising molecular target for anticancer treatment strategies. Notably, FOXM1 also serves as a key mediator of estrogen-induced mitogenic signaling in breast cancer cells [37,38]. We found that RGC-32 regulated FoxM1 expression in breast cancer cells in the presence of ERα. Several known downstream targets of FoxM1 including SKP2 and SOX2 are important for tamoxifen resistance [39,40]. Our results showed that FoxM1 overexpression reversed the inhibitory effects of RGC-32 knockdown on these gene expression and cell proliferation, suggesting that FoxM1 is a downstream target of RGC-32. RGC-32 is physically associated with Cdk1 and increased the kinase activity [6,8]. Cdk1 phosphorylates FoxM1, resulting in the activation of FoxM1, which is indispensable for regulating the expression of diverse mitotic genes [41]. Our results showed that the kinase activity of Cdk1 was regulated by RGC-32 in ER⁺ breast cancer cells. Inhibition of Cdk1 reduced the level of phosphorylated FoxM1 promoted by RGC-32 overexpression. These results indicate that RGC-32 affects the activation of ERα, thereby promoting the transcriptional activation of FOXM1, thus affecting the cell proliferation of ER+ breast cancer and tamoxifen resistance (Fig 6E).

In summary, RGC-32 is involved in cell proliferation and tamoxifen resistance in ER⁺ breast cancer, which can potentially be employed as a prognostic biomarker for the progression of ER⁺ breast cancer, as well as a predictive biomarker for tamoxifen resistance. Therefore RGC-32 may be a novel target to reverse such resistance.

## Materials and methods

### Cell culture

MCF-7 cells were cultured in DMEM medium with 10% fetal bovine serum (FBS), while T-47D were cultured in RPMI 1640 with 10% FBS. Their identities were confirmed by short tandem repeat (STR) analysis. 17β-estradiol (E2) were purchased from sigma (St. Louis, USA) and dissolved in ethanol. Tamoxifen and fulvestrant (ICI182,780) were purchased from MCE.

For hormone starvation of ER$^+$ breast cancer cells, cells were cultured in 5% FBS pretreated with dextran-coated charcoal and phenol red-free DMEM for 6 days or the indicated time.

### Immunohistochemistry (IHC)

Breast cancer tissue specimens were randomly collected who underwent radical resection at the 971 Hospital of People's Liberation Army from 11/01/2013–27/12/2017. The included breast cancer specimens were pathologically confirmed. Written informed consent for this study was obtained from each patient and the protocol was approved by the Ethics Committee of Qingdao Central Hospital. IHC was performed in 11/07/2024 as described previously [42]. Specimens were incubated with appropriate dilutions of primary antibodies RGC-32 (Abcam, ab221098) and FOXM1 (Invitrogen, 702664) and Ki67 (Abcam, ab15580), and then incubated with second antibodies at 37 °C for 1 hour. IHC staining for RGC-32 and FoxM1 was independently reviewed and scored by two pathologists using the Allred score. Ki-67 immunoreactivity was recorded as continuous variables, based on the proportion of positive tumor cells (0–100%) regardless of staining intensity.

### Plasmid construct

For RNA interference, short hairpin RNAs (shRNA) targeting RGC-32 and Cdk1 were cloned into lentivirus vector GV248 (Genechem, Shanghai, China) and pGPU6 vector (GenePharma, Shanghai, China), respectively. Scrambled shRNA sequence was used as a control. All nucleotide sequences used in this study are described in S1 Table. The coding sequence for RGC-32 and FoxM1 were cloned into the pcDNA3.0 vector and LV5 vector (GenePharma, Shanghai, China), respectively.

### Luciferase reporter assays

The estrogen signaling luciferase activity was measured as described [43]. Breast cells were transfected with the ERE luciferase reporter along with Renilla using X-tremegene HP (ROCHE). The activity of luciferase was measured 24 h post-treatment.

### Cell proliferation assay

For cell counting kit-8 (CCK8) assay, breast cancer cells were plated in 96-well plates and treated with various chemotherapeutic agents for the indicated time. The cells were incubated with 10μl/well of CCK-8 solution for 2h. The absorbance at 450 nm of each well was read on a spectrophotometer.

### Cell cycle assay

Breast cancer cells were fixed in 75% ethanal overnight at 4°C. Then cells were stained with 10 μg/ml PI in PBS plus RNase at 37°C for 30 min. The cells were analyzed by a flow cytometer.

### Immunoblot analysis

Breast cancer cells were lysed in RIPA buffer containing a protease inhibitor cocktail and quantified by Bradford reagent. Proteins were separated by SDS-PAGE, and then transferred to polyvinylidene difluoride membranes. The membranes were blocked with 5% non-fat milk, then and immunoblotted with antibodies: ERα, p-ERα (Abcam, ab31478), RGC-32 (Invitrogen, PA5–120573), FoxM1 (Invitrogen, 702664), phospho-FoxM1 at Thr600 and Ser35 (Invitrogen, PA5–105625, PA5–105197), Cdk1 (Abcam, ab265590), total-AKT (Cell signaling, 4697) and p-AKT (Cell signaling, 4060). HRP-conjugated anti-rabbit were used as secondary antibodies. The protein signals were detected with enhanced chemiluminescence.

## Microarray analysis

Total RNA extracted from MCF-7 cells transfected with either RGC-32 shRNA or a scrambled shRNA vector was used in the Affymetrix cDNA microarray analysis. In the analysis, hybridization was performed with the GeneChip Human Clariom S Array (Affymetrix, Santa Clara, CA), and the chips were scanned with an Affymetrix GeneChip scanner 3000. Data files were generated and processed with Affymetrix software and analyzed with Transcriptome Analysis Console Software v. 3.0 (all from Thermo Fisher Scientific, Waltham, MA). Differential gene expression between experimental groups was assessed using one-way ANOVA, with subsequent pathway enrichment analysis conducted through IPA (Qiagen Sciences, Germantown, MD).

## Quantitative RT-PCR

Total RNAs from indicated cells was extracted using the TRIZOL reagent (Invitrogen) according to the manufacturer's instructions. cDNA was synthesized using HiScript II Q RT SuperMix (Vazyme, Jiangsu, China), and then subjected to qRT-PCR using ChamQ Universal SYBR qPCR Master Mix (Vazyme). The comparative threshold cycle (CT) method was used to calculate the relative expression of specific genes with proper primers. The GAPDH was used as an internal control.

## Tumor xenograft

Briefly, $1 \times 10^6$ MCF-7 cells transfected with RGC-32 or a control vector were injected into the mammary fat pads of mice in the orthotopic model. E2 pellet containing 0.18 mg of 17β-estradiol for slow-release and/or another pellet containing 5 mg tamoxifen (Innovative Research of America) was subcutaneously implanted in the lateral region of the neck of nude mice. Tumor size was determined by caliper measurements every three days and tumor volumes were calculated as $0.5 \times length \times width^2$. Mice were sacrificed after 5 weeks and the primary mammary tumors were collected for histological analysis.

## Statistical analysis

All data was expressed as the mean ± SD of three independent experiments. Comparison of two or multiple groups was analyzed using Student's t-test or one-way analysis of variance (ANOVA). Statistical analysis of tissue samples was performed using the $\chi^2$ test, whereas the survival analysis was performed by Kaplan-Meier analysis and log-rank test. The correlation of protein expression of RGC-32 and FoxM1 was analyzed by linear regression analysis. Multivariate analyses were executed via Cox proportional hazard model.

## Supporting information

**S1 Fig. RGC-32 expression in ER$^+$ breast cancer.** A-B, The expression levels of RGC-32 in breast cancer cells stably transfected with RGC-32 shRNA and control shRNA or plasmids containing RGC-32 and the empty vector, were determined by western-blotting. C, Parental and tamoxifen resistant (TamR) cells were treated with 1μM tamoxifen or vehicle for 5 days. Cell viability was assessed by CCK8 assay. D, The expression levels of RGC-32 in parental and TamR cells. E, The expression levels of RGC-32 in TamR cells stably transfected with RGC-32 shRNA and control shRNA were determined by western-blotting.
(TIF)

**S2 Fig. FoxM1 reversed the inhibitory effects of RGC-32 knockdown on SKP2 and SOX2 mRNA expression.** A-B, RGC-32 knockdown decreased SKP2 and SOX2 mRNA expression in MCF-7 and T-47D cells. *$P < 0.05$; **$P < 0.01$; ***$P < 0.001$ for target gene expression comparison. C, Breast cancer cells were transfected with plasmids expressing

FoxM1. The expression levels of FoxM1 were determined by western-blotting. D, Breast cancer cells were co-transfected with plasmids expressing FoxM1, RGC-32-shRNA or empty vector. SKP2 and SOX2 mRNA expression were determined by Q-PCR.
(TIF)

**S1 Table. Sequences of the shRNAs and primers for ChIP assays and qRT-PCR analysis.**
(DOCX)

**S1 File. Original images for blots and gels.**
(PDF)

## Author contributions

**Conceptualization:** Xinlei Li, Wei Zhang.

**Data curation:** Yan Liu, Zhiqian Wang, Yu Wang.

**Formal analysis:** Xiaocui Bu.

**Funding acquisition:** Peng Zhao.

**Investigation:** Xinlei Li, Yan Liu, Zhiqian Wang, Xiaocui Bu.

**Project administration:** Peng Zhao.

**Resources:** Yan Liu, Wei Zhang.

**Supervision:** Peng Zhao.

**Validation:** Peng Zhao, Wei Zhang.

**Writing – original draft:** Xinlei Li.

**Writing – review & editing:** Peng Zhao, Wei Zhang.

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
