## [Decision Letter · Decision Letter 0]

Dear Dr. Zhao,

Thank you for submitting your manuscript to PLOS ONE. After careful consideration, we feel that it has merit but does not fully meet PLOS ONE’s publication criteria as it currently stands. Therefore, we invite you to submit a revised version of the manuscript that addresses the points raised during the review process.

We look forward to receiving your revised manuscript.

Kind regards,

Tudor C. Badea, M.D., M.A., Ph.D.

Academic Editor

PLOS ONE

Journal Requirements:

“This work was supported by a grant from the National Natural Science Foundation of China (NO. 82072927).”

4. We note that your Data Availability Statement is currently as follows: All relevant data are within the manuscript and in Supporting Information files.

7. PLOS ONE now requires that authors provide the original uncropped and unadjusted images underlying all blot or gel results reported in a submission’s figures or Supporting Information files. This policy and the journal’s other requirements for blot/gel reporting and figure preparation are described in detail at https://journals.plos.org/plosone/s/figures#loc-blot-and-gel-reporting-requirements and https://journals.plos.org/plosone/s/figures#loc-preparing-figures-from-image-files. When you submit your revised manuscript, please ensure that your figures adhere fully to these guidelines and provide the original underlying images for all blot or gel data reported in your submission. See the following link for instructions on providing the original image data: https://journals.plos.org/plosone/s/figures#loc-original-images-for-blots-and-gels.  

Additional Editor Comments:

The reviewers made suggestions for improving your manuscript, in particular related to the placing in context of your findings, i.e. introduction and discussion, and data processing of microarray data.

This last point is particularly important, as availability of microarray data published in this study is mandatory before considering publication.

Please deposit the microarray data in a public repository and describe the format, accession number and release date of the data in a separate paragraph in the data availability section. Also please expand on the performed DEG analysis in both methods and results section.

Reviewers' comments:

Reviewer's Responses to Questions

**Comments to the Author**

1. Is the manuscript technically sound, and do the data support the conclusions?

Reviewer #1: Yes

Reviewer #2: Yes

2. Has the statistical analysis been performed appropriately and rigorously?

Reviewer #1: Yes

Reviewer #2: Yes

3. Have the authors made all data underlying the findings in their manuscript fully available?

Reviewer #1: Yes

Reviewer #2: Yes

4. Is the manuscript presented in an intelligible fashion and written in standard English?

Reviewer #1: Yes

Reviewer #2: Yes

Reviewer #1: This is an interesting manuscript investigating the role of RGC-32 in cell proliferation and tamoxifen resistance in breast cancer. The authors show that the RGC-32 mediated tamoxifen resistance is partially dependent on FoxM1 expression. Targeting RGC-32 can be an effective target in restoring sensitivity to Tamoxifen in breast cancer cells.

Critique

The introduction is very short. Adding more details on the role of RGC-32 in cancers will be helpful for the reader.

The Discussion section will benefit from more details on how RGC_32 activate cell cycle and induce cell proliferation, especially concerning phosphorylation sites induced activated by CDK1, Akt and ERK1.

Discussing the role of ERK1 in the context of cell proliferation will be also helpful.

More details on the role of Fox M1 role in pathology and cancers will also be helpful.

Reviewer #2: Strengths of the study:

This is a very thorough translational and experimental research study that eloquently documents RGC-32 expression in ER+ breast cancer tissues, while also deciphering the mechanisms through which RGC-32 mediates tamoxifen resistance and promotes breast cancer cells cell proliferation.

The methods used in this study are well documented and valid: immunohistochemistry, RNA interference, microarray gene analysis and the use of tumor xenograft model.

The authors demonstrate here that RGC-32 can activate PI3K signaling pathway, consequently enhancing ERα activity. ERα activation is essential for RGC-32-mediated the expression of FoxM1, while Tamoxifen resistance mediated by RGC-32 seems partially dependent on FoxM1 expression.

We believe that the data presented here, an elucidation of the molecular

mechanisms of acquired drug resistance in breast cancer, represents a real progress in the field of translational oncology.

Minor issues (limitations):

1. Abstract, Page 10: the acronym ERα is not detailed.

2. Neither in the DISCUSSION section, nor in the Introduction section you do not mention the connection with the complement system, namely with C5b-9, knowing that RGC-32 is a protein that promotes cell cycle progression in response to complement activation (Badea et al, J Biol Chem 2002).

Please detail this connection and integrate it in harmony with the data presented here (your results).

3. In the Discussion section, when discussing the role played by RGC-32 in the proliferation of ER+ breast cancer cells, other immunohistochemistry data on RGC-32’s expression in cancer tissue ought to be mentioned: we have now shown that RGC-32 is expressed in precancerous states, and its expression is significantly higher in adenomas than in normal colon tissue. The expression of RGC-32 was higher in advanced stages of colon cancer than in precancerous states or the initial stages of colon cancer. (S.I. Vlaicu et al. Experimental and Molecular Pathology 88 (2010) 67–76.)

4. The gene array data you performed in MCF-7 cells is insufficiently exploited in my view: what are other relevant pathways influenced by RGC-32 silencing, apart cell cycle control of chromosomal replication? And why is there no reference of microarray data that precedes your work: a gene array analysis to investigate the effect of RGC-32 knockdown on gene expression in the SW480 colon cancer cell line S.I. Vlaicu et al. Experimental and Molecular Pathology 88 (2010) 67–76.).

A comparison the different RGC-32 dependent pathways found in these two studies should be present in your Discussion section.

5. You mention that “These results suggested a novel regulatory relationship between RGC-32 and PI3K/AKT-ERα crosstalk.” I would disagree that this relationship is a NEW finding, and here are the grounds for my disapproval:

This relationship was previously documented in endothelial cells by Fosbrink et al. (Fosbrink et al., Exp Mol Pahol Vol 86, 2009) and in T-lymphocytes by Tegla et al. (Tegla CA, Cudrici CD, Nguyen V, Danoff J, Kruszewski AM, Boodhoo D, Mekala AP, Vlaicu SI, Chen C, Rus V, Badea TC, Rus H. RGC-32 is a novel regulator of the T-lymphocyte cell cycle. Exp Mol Pathol. 2015 Jun;98(3):328-37.) The aforementioned research showed that RGC-32 was involved in controlling the cell cycle of T cells in vivo; AKT connects PI3K to signaling pathways that promote cytokine transcription, survival, and cell-cycle activation. RGC-32 acts on Akt which phosphorylates FOXO1 protein, induces IL-2 transcription and cell cycle activation contributing to TCR mediated signaling.

I would see it as optimal and necessary to integrate these previous findings into the Discussion chapter.

My recommendation: Accepted for publication with minor revisions.

**Do you want your identity to be public for this peer review?** For information about this choice, including consent withdrawal, please see our Privacy Policy

Reviewer #1: No

Reviewer #2: **Yes: ** Sonia Irina Vlaicu

---

## [Author Response · Author response to Decision Letter 1]

8 May 2025

Reviewer #1: This is an interesting manuscript investigating the role of RGC-32 in cell proliferation and tamoxifen resistance in breast cancer. The authors show that the RGC-32 mediated tamoxifen resistance is partially dependent on FoxM1 expression. Targeting RGC-32 can be an effective target in restoring sensitivity to Tamoxifen in breast cancer cells.

We would like to express our sincere thanks to the reviewers for the constructive and positive comments.

Critique

The introduction is very short. Adding more details on the role of RGC-32 in cancers will be helpful for the reader.

Answer: We expand the introduction (page 40, paragraph 2) to include the more details on the role of RGC-32 in cancers.

The Discussion section will benefit from more details on how RGC_32 activate cell cycle and induce cell proliferation, especially concerning phosphorylation sites induced activated by CDK1, Akt and ERK1.

Answer: Several sentences have been added in the Discussion (page 45, paragraph 4) in the revised version to address this issue.

Discussing the role of ERK1 in the context of cell proliferation will be also helpful.

Answer: we added the role of ERK1 in cell proliferation in the Discussion (page 45, paragraph 4) in the revised version.

More details on the role of Fox M1 role in pathology and cancers will also be helpful.

Answer: Several sentences have been added in the Discussion (page 46, paragraph 2) in the revised version to address this issue.

Reviewer #2: Strengths of the study:

This is a very thorough translational and experimental research study that eloquently documents RGC-32 expression in ER+ breast cancer tissues, while also deciphering the mechanisms through which RGC-32 mediates tamoxifen resistance and promotes breast cancer cells cell proliferation.

The methods used in this study are well documented and valid: immunohistochemistry, RNA interference, microarray gene analysis and the use of tumor xenograft model.

The authors demonstrate here that RGC-32 can activate PI3K signaling pathway, consequently enhancing ERα activity. ERα activation is essential for RGC-32-mediated the expression of FoxM1, while Tamoxifen resistance mediated by RGC-32 seems partially dependent on FoxM1 expression.

We believe that the data presented here, an elucidation of the molecular mechanisms of acquired drug resistance in breast cancer, represents a real progress in the field of translational oncology.

We would like to express our sincere thanks to the reviewers for the constructive and positive comments.

Minor issues (limitations):

1. Abstract, Page 10: the acronym ERα is not detailed.1.

Answer: Correction has been made in the revised version.

2. Neither in the DISCUSSION section, nor in the Introduction section you do not mention the connection with the complement system, namely with C5b-9, knowing that RGC-32 is a protein that promotes cell cycle progression in response to complement activation (Badea et al, J Biol Chem 2002).

Please detail this connection and integrate it in harmony with the data presented here (your results).

Answer: Several sentences have been added in the Introduction (page 40, paragraph 2) in the revised version to address this issue.

3. In the Discussion section, when discussing the role played by RGC-32 in the proliferation of ER+ breast cancer cells, other immunohistochemistry data on RGC-32’s expression in cancer tissue ought to be mentioned: we have now shown that RGC-32 is expressed in precancerous states, and its expression is significantly higher in adenomas than in normal colon tissue. The expression of RGC-32 was higher in advanced stages of colon cancer than in precancerous states or the initial stages of colon cancer. (S.I. Vlaicu et al. Experimental and Molecular Pathology 88 (2010) 67–76.)

Answer: Several sentences have been added in the Discussion (page 46, paragraph 3) in the revised version to address this issue.

4. The gene array data you performed in MCF-7 cells is insufficiently exploited in my view: what are other relevant pathways influenced by RGC-32 silencing, apart cell cycle control of chromosomal replication? And why is there no reference of microarray data that precedes your work: a gene array analysis to investigate the effect of RGC-32 knockdown on gene expression in the SW480 colon cancer cell line S.I. Vlaicu et al. Experimental and Molecular Pathology 88 (2010) 67–76.).

A comparison the different RGC-32 dependent pathways found in these two studies should be present in your Discussion section.

Answer: Several sentences have been added in the Discussion (page 46, paragraph 3) in the revised version to address this issue.

5. You mention that “These results suggested a novel regulatory relationship between RGC-32 and PI3K/AKT-ERα crosstalk.” I would disagree that this relationship is a NEW finding, and here are the grounds for my disapproval:

This relationship was previously documented in endothelial cells by Fosbrink et al. (Fosbrink et al., Exp Mol Pahol Vol 86, 2009) and in T-lymphocytes by Tegla et al. (Tegla CA, Cudrici CD, Nguyen V, Danoff J, Kruszewski AM, Boodhoo D, Mekala AP, Vlaicu SI, Chen C, Rus V, Badea TC, Rus H. RGC-32 is a novel regulator of the T-lymphocyte cell cycle. Exp Mol Pathol. 2015 Jun;98(3):328-37.) The aforementioned research showed that RGC-32 was involved in controlling the cell cycle of T cells in vivo; AKT connects PI3K to signaling pathways that promote cytokine transcription, survival, and cell-cycle activation. RGC-32 acts on Akt which phosphorylates FOXO1 protein, induces IL-2 transcription and cell cycle activation contributing to TCR mediated signaling.

I would see it as optimal and necessary to integrate these previous findings into the Discussion chapter.

Answer: we have integrated these previous findings into the Discussion chapter (page 46, paragraph 4).

---

## [Editor Report · Decision Letter 1]

Dear Dr. Zhao,

Thank you for submitting your manuscript to PLOS ONE. After careful consideration, we feel that it has merit but does not fully meet PLOS ONE’s publication criteria as it currently stands. Therefore, we invite you to submit a revised version of the manuscript that addresses the points raised during the review process.

Please comply with data sharing requirements, by making microarray data publicly available as indicated.

We look forward to receiving your revised manuscript.

Kind regards,

Tudor C. Badea, M.D., M.A., Ph.D.

Academic Editor

PLOS ONE

Additional Editor Comments:

The results of microarray experiments are not sufficiently presented. The raw counts should be submitted to a repository (e.g. gene expression omnibus), and the accession number provided and supplementary tables with differrentially expressed genes should be provided.

Making your data available in this fashion is mandatory for publication.

---

## [Author Response · Author response to Decision Letter 2]

13 May 2025

Reviewer #1: This is an interesting manuscript investigating the role of RGC-32 in cell proliferation and tamoxifen resistance in breast cancer. The authors show that the RGC-32 mediated tamoxifen resistance is partially dependent on FoxM1 expression. Targeting RGC-32 can be an effective target in restoring sensitivity to Tamoxifen in breast cancer cells.

We would like to express our sincere thanks to the reviewers for the constructive and positive comments.

Critique

The introduction is very short. Adding more details on the role of RGC-32 in cancers will be helpful for the reader.

Answer: We expand the introduction (page 40, paragraph 2) to include the more details on the role of RGC-32 in cancers.

The Discussion section will benefit from more details on how RGC_32 activate cell cycle and induce cell proliferation, especially concerning phosphorylation sites induced activated by CDK1, Akt and ERK1.

Answer: Several sentences have been added in the Discussion (page 45, paragraph 4) in the revised version to address this issue.

Discussing the role of ERK1 in the context of cell proliferation will be also helpful.

Answer: we added the role of ERK1 in cell proliferation in the Discussion (page 45, paragraph 4) in the revised version.

More details on the role of Fox M1 role in pathology and cancers will also be helpful.

Answer: Several sentences have been added in the Discussion (page 46, paragraph 2) in the revised version to address this issue.

Reviewer #2: Strengths of the study:

This is a very thorough translational and experimental research study that eloquently documents RGC-32 expression in ER+ breast cancer tissues, while also deciphering the mechanisms through which RGC-32 mediates tamoxifen resistance and promotes breast cancer cells cell proliferation.

The methods used in this study are well documented and valid: immunohistochemistry, RNA interference, microarray gene analysis and the use of tumor xenograft model.

The authors demonstrate here that RGC-32 can activate PI3K signaling pathway, consequently enhancing ERα activity. ERα activation is essential for RGC-32-mediated the expression of FoxM1, while Tamoxifen resistance mediated by RGC-32 seems partially dependent on FoxM1 expression.

We believe that the data presented here, an elucidation of the molecular mechanisms of acquired drug resistance in breast cancer, represents a real progress in the field of translational oncology.

We would like to express our sincere thanks to the reviewers for the constructive and positive comments.

Minor issues (limitations):

1. Abstract, Page 10: the acronym ERα is not detailed.1.

Answer: Correction has been made in the revised version.

2. Neither in the DISCUSSION section, nor in the Introduction section you do not mention the connection with the complement system, namely with C5b-9, knowing that RGC-32 is a protein that promotes cell cycle progression in response to complement activation (Badea et al, J Biol Chem 2002).

Please detail this connection and integrate it in harmony with the data presented here (your results).

Answer: Several sentences have been added in the Introduction (page 40, paragraph 2) in the revised version to address this issue.

3. In the Discussion section, when discussing the role played by RGC-32 in the proliferation of ER+ breast cancer cells, other immunohistochemistry data on RGC-32’s expression in cancer tissue ought to be mentioned: we have now shown that RGC-32 is expressed in precancerous states, and its expression is significantly higher in adenomas than in normal colon tissue. The expression of RGC-32 was higher in advanced stages of colon cancer than in precancerous states or the initial stages of colon cancer. (S.I. Vlaicu et al. Experimental and Molecular Pathology 88 (2010) 67–76.)

Answer: Several sentences have been added in the Discussion (page 46, paragraph 3) in the revised version to address this issue.

4. The gene array data you performed in MCF-7 cells is insufficiently exploited in my view: what are other relevant pathways influenced by RGC-32 silencing, apart cell cycle control of chromosomal replication? And why is there no reference of microarray data that precedes your work: a gene array analysis to investigate the effect of RGC-32 knockdown on gene expression in the SW480 colon cancer cell line S.I. Vlaicu et al. Experimental and Molecular Pathology 88 (2010) 67–76.).

A comparison the different RGC-32 dependent pathways found in these two studies should be present in your Discussion section.

Answer: Several sentences have been added in the Discussion (page 46, paragraph 3) in the revised version to address this issue.

5. You mention that “These results suggested a novel regulatory relationship between RGC-32 and PI3K/AKT-ERα crosstalk.” I would disagree that this relationship is a NEW finding, and here are the grounds for my disapproval:

This relationship was previously documented in endothelial cells by Fosbrink et al. (Fosbrink et al., Exp Mol Pahol Vol 86, 2009) and in T-lymphocytes by Tegla et al. (Tegla CA, Cudrici CD, Nguyen V, Danoff J, Kruszewski AM, Boodhoo D, Mekala AP, Vlaicu SI, Chen C, Rus V, Badea TC, Rus H. RGC-32 is a novel regulator of the T-lymphocyte cell cycle. Exp Mol Pathol. 2015 Jun;98(3):328-37.) The aforementioned research showed that RGC-32 was involved in controlling the cell cycle of T cells in vivo; AKT connects PI3K to signaling pathways that promote cytokine transcription, survival, and cell-cycle activation. RGC-32 acts on Akt which phosphorylates FOXO1 protein, induces IL-2 transcription and cell cycle activation contributing to TCR mediated signaling.

I would see it as optimal and necessary to integrate these previous findings into the Discussion chapter.

Answer: we have integrated these previous findings into the Discussion chapter (page 46, paragraph 4).

Additional Editor Comments:

The reviewers made suggestions for improving your manuscript, in particular related to the placing in context of your findings, i.e. introduction and discussion, and data processing of microarray data.

This last point is particularly important, as availability of microarray data published in this study is mandatory before considering publication.

Please deposit the microarray data in a public repository and describe the format, accession number and release date of the data in a separate paragraph in the data availability section. Also please expand on the performed DEG analysis in both methods and results section.

Answer: The microarray data have been deposited in the ArrayExpress database (accession number E-MTAB-15130) on May 6, 2025. Detailed DEG analysis was provided in methods section.

---

## [Editor Report · Decision Letter 2]

Dear Dr. Zhao,

Thank you for submitting your manuscript to PLOS ONE. After careful consideration, we feel that it has merit but does not fully meet PLOS ONE’s publication criteria as it currently stands. Therefore, we invite you to submit a revised version of the manuscript that addresses the points raised during the review process.

We look forward to receiving your revised manuscript.

Kind regards,

Tudor C. Badea, M.D., M.A., Ph.D.

Academic Editor

PLOS ONE

---

## [Author Response · Author response to Decision Letter 3]

15 Jun 2025

Reviewer #1: This is an interesting manuscript investigating the role of RGC-32 in cell proliferation and tamoxifen resistance in breast cancer. The authors show that the RGC-32 mediated tamoxifen resistance is partially dependent on FoxM1 expression. Targeting RGC-32 can be an effective target in restoring sensitivity to Tamoxifen in breast cancer cells.

We would like to express our sincere thanks to the reviewers for the constructive and positive comments.

Critique

The introduction is very short. Adding more details on the role of RGC-32 in cancers will be helpful for the reader.

Answer: We expand the introduction (page 40, paragraph 2) to include the more details on the role of RGC-32 in cancers.

The Discussion section will benefit from more details on how RGC_32 activate cell cycle and induce cell proliferation, especially concerning phosphorylation sites induced activated by CDK1, Akt and ERK1.

Answer: Several sentences have been added in the Discussion (page 45, paragraph 4) in the revised version to address this issue.

Discussing the role of ERK1 in the context of cell proliferation will be also helpful.

Answer: we added the role of ERK1 in cell proliferation in the Discussion (page 45, paragraph 4) in the revised version.

More details on the role of Fox M1 role in pathology and cancers will also be helpful.

Answer: Several sentences have been added in the Discussion (page 46, paragraph 2) in the revised version to address this issue.

Reviewer #2: Strengths of the study:

This is a very thorough translational and experimental research study that eloquently documents RGC-32 expression in ER+ breast cancer tissues, while also deciphering the mechanisms through which RGC-32 mediates tamoxifen resistance and promotes breast cancer cells cell proliferation.

The methods used in this study are well documented and valid: immunohistochemistry, RNA interference, microarray gene analysis and the use of tumor xenograft model.

The authors demonstrate here that RGC-32 can activate PI3K signaling pathway, consequently enhancing ERα activity. ERα activation is essential for RGC-32-mediated the expression of FoxM1, while Tamoxifen resistance mediated by RGC-32 seems partially dependent on FoxM1 expression.

We believe that the data presented here, an elucidation of the molecular mechanisms of acquired drug resistance in breast cancer, represents a real progress in the field of translational oncology.

We would like to express our sincere thanks to the reviewers for the constructive and positive comments.

Minor issues (limitations):

1. Abstract, Page 10: the acronym ERα is not detailed.1.

Answer: Correction has been made in the revised version.

2. Neither in the DISCUSSION section, nor in the Introduction section you do not mention the connection with the complement system, namely with C5b-9, knowing that RGC-32 is a protein that promotes cell cycle progression in response to complement activation (Badea et al, J Biol Chem 2002).

Please detail this connection and integrate it in harmony with the data presented here (your results).

Answer: Several sentences have been added in the Introduction (page 40, paragraph 2) in the revised version to address this issue.

3. In the Discussion section, when discussing the role played by RGC-32 in the proliferation of ER+ breast cancer cells, other immunohistochemistry data on RGC-32’s expression in cancer tissue ought to be mentioned: we have now shown that RGC-32 is expressed in precancerous states, and its expression is significantly higher in adenomas than in normal colon tissue. The expression of RGC-32 was higher in advanced stages of colon cancer than in precancerous states or the initial stages of colon cancer. (S.I. Vlaicu et al. Experimental and Molecular Pathology 88 (2010) 67–76.)

Answer: Several sentences have been added in the Discussion (page 46, paragraph 3) in the revised version to address this issue.

4. The gene array data you performed in MCF-7 cells is insufficiently exploited in my view: what are other relevant pathways influenced by RGC-32 silencing, apart cell cycle control of chromosomal replication? And why is there no reference of microarray data that precedes your work: a gene array analysis to investigate the effect of RGC-32 knockdown on gene expression in the SW480 colon cancer cell line S.I. Vlaicu et al. Experimental and Molecular Pathology 88 (2010) 67–76.).

A comparison the different RGC-32 dependent pathways found in these two studies should be present in your Discussion section.

Answer: Several sentences have been added in the Discussion (page 46, paragraph 3) in the revised version to address this issue.

5. You mention that “These results suggested a novel regulatory relationship between RGC-32 and PI3K/AKT-ERα crosstalk.” I would disagree that this relationship is a NEW finding, and here are the grounds for my disapproval:

This relationship was previously documented in endothelial cells by Fosbrink et al. (Fosbrink et al., Exp Mol Pahol Vol 86, 2009) and in T-lymphocytes by Tegla et al. (Tegla CA, Cudrici CD, Nguyen V, Danoff J, Kruszewski AM, Boodhoo D, Mekala AP, Vlaicu SI, Chen C, Rus V, Badea TC, Rus H. RGC-32 is a novel regulator of the T-lymphocyte cell cycle. Exp Mol Pathol. 2015 Jun;98(3):328-37.) The aforementioned research showed that RGC-32 was involved in controlling the cell cycle of T cells in vivo; AKT connects PI3K to signaling pathways that promote cytokine transcription, survival, and cell-cycle activation. RGC-32 acts on Akt which phosphorylates FOXO1 protein, induces IL-2 transcription and cell cycle activation contributing to TCR mediated signaling.

I would see it as optimal and necessary to integrate these previous findings into the Discussion chapter.

Answer: we have integrated these previous findings into the Discussion chapter (page 46, paragraph 4).

Additional Editor Comments:

The reviewers made suggestions for improving your manuscript, in particular related to the placing in context of your findings, i.e. introduction and discussion, and data processing of microarray data.

This last point is particularly important, as availability of microarray data published in this study is mandatory before considering publication.

Please deposit the microarray data in a public repository and describe the format, accession number and release date of the data in a separate paragraph in the data availability section. Also please expand on the performed DEG analysis in both methods and results section.

Answer: The microarray data have been deposited in the ArrayExpress database (accession number E-MTAB-15130), made publicly available on June 6, 2025.

---

## [Editor Report · Decision Letter 3]

Response Gene to Complement 32 promotes cell proliferation and tamoxifen resistance in breast cancer via elevated FoxM1 expression

PONE-D-25-01923R3

Dear Dr. Zhao,

We’re pleased to inform you that your manuscript has been judged scientifically suitable for publication and will be formally accepted for publication once it meets all outstanding technical requirements.

Kind regards,

Tudor C. Badea, M.D., M.A., Ph.D.

Academic Editor

PLOS ONE
---

## [Editor Report · Acceptance letter]

PONE-D-25-01923R3

PLOS ONE

Dear Dr. Zhao,

I'm pleased to inform you that your manuscript has been deemed suitable for publication in PLOS ONE. Congratulations! Your manuscript is now being handed over to our production team.

Kind regards,

on behalf of

Dr. Tudor C. Badea

Academic Editor

PLOS ONE